# Equine Nuclear Medicine in 2024: Use and Value of Scintigraphy and PET in Equine Lameness Diagnosis

**DOI:** 10.3390/ani14172499

**Published:** 2024-08-28

**Authors:** Mathieu Spriet, Filip Vandenberghe

**Affiliations:** 1Department of Surgical and Radiological Sciences, School of Veterinary Medicine, University of California, Davis, CA 95616, USA; 2Equine Hospital Bosdreef, 9180 Moerbeke-Waas, Belgium

**Keywords:** horse, imaging, bone, joint, tendon, Technetium-99m, 18F-Sodium Fluoride, 18F-Fluorodeoxyglucose

## Abstract

**Simple Summary:**

Nuclear medicine imaging techniques consist of acquiring images after intravenous administration of a small dose of radioactive molecules. Different nuclear medicine techniques are available in the horse. Scintigraphy provides two-dimensional images and has been used for the detection of bone injuries in horses since the late 1970s. This was at the time the only alternative to radiographs to detect bone issues in horses. As other imaging techniques such as computed tomography (CT) and magnetic resonance imaging (MRI) have developed, the role of scintigraphy in horses has reduced. Scintigraphy remains, however, commonly used in racehorses to detect stress injuries and is also used in other horses to image larger body areas or even the whole body of a horse. In the last 10 years, positron emission tomography (PET) has become available in horses and provides higher resolution three-dimensional images. PET shows promising applications by either replacing the use of scintigraphy for the smaller areas of the limbs or by adding useful information to other imaging techniques. PET allows detection of some injuries prior to their identification with CT or MRI and also helps decide if an injury is active or represents an old scar.

**Abstract:**

Scintigraphy and Positron Emission Tomography (PET) are both nuclear medicine imaging techniques, providing functional information of the imaged areas. Scintigraphy is a two-dimensional projected imaging technique that was introduced in equine imaging in the late 1970s. Scintigraphy allows imaging of large body parts and can cover multiple areas, remaining the only technique commonly used in horses for whole body imaging. PET is a cross-sectional imaging technique, first used in horses in 2015, allowing higher resolution three-dimensional functional imaging of the equine distal limb. This manuscript will cover current use and values of these two modalities in equine lameness diagnosis.

## 1. Introduction

Nuclear medicine imaging techniques rely on the biodistribution of a radiopharmaceutical agent to create an image, providing functional information of the imaged area, rather than structural information as classically reported by other imaging techniques. Depending on the radiotracer used, different physiological processes can be imaged. Scintigraphy in horses is primarily used for bone imaging. The main radiopharmaceutical used for bone scintigraphy is Technetium-99m (^99m^Tc) coupled with a bisphosphonates agent. Scintigraphy was first introduced for equine imaging by Ueltschi in Bern, Switzerland in the late 1970s [1] and was at the time the only available imaging alternative to radiography. The use of scintigraphy in horses increased in the 1980s and 1990s [2,3,4], but the development of other advanced imaging modalities such as Computed Tomography (CT) and Magnetic Resonance Imaging (MRI) in the 2000s resulted in a reduction in the use of scintigraphy for distal limb imaging. Nevertheless, in 2023, scintigraphy is still the only imaging technique routinely used to perform whole body imaging and remains the reference imaging tool for some specific indications such as stress fractures of the proximal limb in racehorses. Scintigraphy also maintained an important role for vertebral column and pelvis imaging in sport horses. Scintigraphy also still has a role in combination with CT and MRI for further characterization of some lesions.

Positron Emission Tomography (PET) is the latest addition to the equine diagnostic imaging arsenal. PET was first performed on a horse at the University of California Davis in 2015 [5]. The first scans were performed under anesthesia using a PET scanner designed to image the human brain. A major advancement for equine PET was the development in 2019 of an equine-specific PET scanner allowing imaging of the distal limb on standing sedated horses [6]. As of mid-2023, ten of these scanners were in use in major equine hospitals in the USA, both in academia and private practices. Two different tracers are commonly used for equine PET. 18F-Sodium Fluoride (18F-NaF) is a bone tracer allowing acquisition of 3D bone scans [7]. 18F-Fluorodeoxyglucose (18F-FDG) is a marker of metabolic activity of glucose, commonly used in oncology, but also pertinent for imaging of inflammation in soft tissues such as tendons, ligaments, and hoof laminae [5].

The objectives of this manuscript are to present the clinical use and value of scintigraphy and PET in 2024. In particular, the aim is to illustrate based on the authors’ clinical experience how the role of scintigraphy has evolved with the development of computed tomography (CT) and magnetic resonance imaging (MRI). The authors also present the new opportunities arising with the recent development of equine PET. This manuscript is not intended to present an exhaustive review of the use of scintigraphy, for which other publications are available [8,9,10].

## 2. Original Indications for Scintigraphy

When scintigraphy was first introduced in the 1970s, it was the only alternative to radiography for equine imaging. This explains why some of the original indications now appear outdated as they have been replaced with other imaging modalities. Original indications for bone scintigraphy included imaging of a focal area when pain has been localized but radiographs do not show any abnormalities, imaging of a large area when lameness cannot be localized with diagnostic analgesia, and whole body imaging for multiple limb lameness or poor performance.

### 2.1. Imaging of a Focal Area When Pain Has Been Localized but Radiographs Do Not Show Any Abnormalities

An early example of this was the imaging of the tarsus when radiographs were inconclusive [11]. This indication has become less common as other imaging alternative may be more useful. For example, CT or MRI would be nowadays the imaging modalities of choice to assess a tarsus for suspicion of a central tarsal bone fracture [12,13]. Another classic indication was the racehorse fetlock as scintigraphy helped recognize and characterize remodeling of the palmar/plantar metacarpal/metatarsal condyles [14,15,16]. These studies have demonstrated that scintigraphy is more sensitive than radiographs to detect stress remodeling [14]. Differences between front and hind limbs have been demonstrated with increased uptake more common medially in the front limbs and laterally in the hind limbs [14,16]. The scintigraphic findings were associated with performance in the racehorses, confirming the value of this modality in clinical practice [16]. However, in the last 10 to 15 years, the development of the cross-sectional imaging techniques has provided opportunities for better characterization of racehorse fetlock lesions with MRI [17,18,19,20], CT [17,18,21,22], or PET [23,24,25], rendering the application of scintigraphy of limited value when the clinical issue has been localized to the fetlocks and other modalities are avaialble.

Vascular and soft tissue phase scintigraphy performed respectively in the first seconds or minutes after injection of the radiopharmaceutical [2,10,26], in comparison with bone scintigraphy typically obtained 2 to 3 h post injection, have had a few specific indications that now have greatly been replaced by other modalities. Vascular scintigraphy allowed diagnosis of aortoiliac thromboembolism [27] but ultrasound has now been developed as a robust and convenient alternative [28,29,30]. Soft tissue phase scintigraphy has been reported for suspensory ligament [2] and deep digital flexor tendon [31] imaging but this has now been by far replaced by ultrasound and/or MRI. A comparative study with MRI regarding identification of deep digital flexor tendon injury in the pastern demonstrated a limited sensitivity (40%) but a high specificity as no false positive cases were identified [31]. Also, lesions were more likely to be detected with scintigraphy on horses with lameness of shorter duration suggesting a higher sensitivity for acute or ongoing injuries [31].

### 2.2. Imaging of a Large Area When Lameness Cannot Be Localized with Diagnostic Analgesia

This indication is extremely valuable in racehorses where lameness might be quite transient and lesions may also involve the proximal limb, rendering localization with diagnostic analgesia more challenging. The nature of lesions encountered in this population with frequent stress remodeling injuries with active bone changes with minimal structural changes make scintigraphy particularly useful [8,14,32,33,34]. Scintigraphy remains the most pertinent imaging modality to detect stress fractures of the humerus [35], tibia or pelvis [36,37]. This indication can also be used in sport horses; however, the interpretation of the scintigraphic scans tend to be more challenging as increased uptake is commonly observed in the spinous processes of the thoracolumbar vertebrae, the tarsi and the proximal phalanges in jumpers, without associated clinical significance [38]. Sport and leisure horses typically present lower uptake due to the older age of the horses and the type of encountered lesions, rendering the positive yield of scintigraphy lower than in racehorses. Degenerative changes more commonly encountered in this population tend to have lower uptake when compared with stress remodeling changes [39]. Scintigraphy was found to have a high specificity (94.0%) but a low sensitivity (43.8%) to identify the cause of the lameness in sports horses, suggesting it should not be used as an isolated or indiscriminate tool for assessment of lameness or poor performance [40]. Low sensitivity related in particular to the detection of proximal suspensory desmopathy, both in front and hind limbs, but the specificity was excellent [40]. Scintigraphy remains, however, particularly pertinent when the distal limb has been ruled out as the source of pain through diagnostic analgesia and imaging of the upper limb is needed. Lesion of the coxofemoral or sacroiliac joints for example do benefit from the ability of scintigraphy to image through thicker body parts [38]; however, the frequent occurrence of false negative results needs to be kept in mind [41].

### 2.3. Whole Body Imaging for Multiple Limb Lameness or Poor Performance

Scintigraphy remains the only imaging modality offering the ability to perform whole body imaging [10]. A whole body scan typically includes 50 or more images and is a time-consuming endeavor that should only be considered in selected cases. This indication is again more likely to be helpful in racehorses than sport horses due to the type of expected injuries. Nevertheless, combined with proper clinical evaluation, scintigraphy continues to be valuable and even regains interest in the evaluation of poor performance in sport horses. It is important to not substitute a whole body bone scan for a proper lameness examination, but acknowledging the risk for false negatives and understanding the occurrence of false positives, whole body scintigraphy remains a useful tool in complex lameness cases [39]. When compared with final diagnosis based on clinical examination and other imaging tests, scintigraphy appears to perform better for feet and proximal aspect of the forelimb (elbow, shoulder and scapula) [40]. It remains important to keep in mind that increased uptake does not necessarily indicate a source of pain as it may also indicate sites of physiologic stress remodeling, clinically silent.

## 3. Roles of Scintigraphy among Other Advanced Imaging Techniques

Modern large equine hospitals are now equipped with multiple diagnostic imaging modalities with various types of computed tomography (CT) and magnetic resonance (MRI) systems now commonly available in the same practice. Each technique having its own strength and limitations, it appears that a combination of different modalities provides the most comprehensive approach to complex lameness issues.

### 3.1. Imaging of Vertebral Column, Ribs and Proximal Limbs

Scintigraphy remains the primary tool for imaging the entire vertebral column. Correlation between radiography and clinical signs has been described to be rather low. The combination of radiographic and scintigraphic findings provides a stronger association with clinical signs [42,43]. CT is increasingly used to examine the cervical and cranial thoracic region but remains very limited for assessment of the thoracic, lumbar, and sacral areas. For cervical imaging, the degree of increased uptake, the possible left–right asymmetry, the abnormal shape or enlargement of articular processes are all pertinent for assessment of cervical joint disease. In the lumbar spine, lateral oblique images provide an opportunity to recognize abnormal uptake associated with the articular processes joints. (Figure 1). Left–right asymmetry in uptake in the lumbar articular processes is an important parameter to consider during the assessment of asymmetrical gait or work.

Scintigraphy was identified as the most sensitive technique to detect rib fractures responsible for performance issues in adult horses [44]. Scintigraphy also allowed the identification of synostosis between the first and second ribs potentially associated with front limb lameness [45].

In addition to imaging of the axial skeleton, scintigraphy remains pertinent for imaging of the joints of the proximal limb in particular shoulder and hip that remain more challenging to assess radiographically due to the large size of surrounding muscle masses.

Scintigraphy also remains a very useful imaging modality to identify myopathies of proximal limbs muscles, in particular the gluteal muscles, caudal thigh muscles and triceps [46,47] (Figure 2). Especially in cases with uptake limited to a certain muscle or muscle bundles, muscle damage is often not large enough to provoke elevated levels of muscle enzymes in the blood and diagnosis is facilitated by scintigraphy and ultrasound with possible guided muscle biopsies. Correlation with clinical signs and recent exercise history remains important, as increased muscular uptake may be an incidental finding in some horses.

### 3.2. Scintigraphy Combined with CT and MRI

The association of scintigraphy with CT and MRI can go both ways, with scintigraphy performed prior or after other advanced modalities. In the active adult equine athlete, a whole body scintigraphy provides an overview of all the conditions that might be present in a horse. This can serve as a baseline in the decision making of which areas may require further imaging. Scintigraphy can, for example, detect increased uptake in the region of the costotransverse joints of the first and second thoracic vertebrae providing an indication for further assessment with CT (Figure 3). Increased uptake can be associated with increased fluid signal in bone (e.g., STIR hyperintensity) and/or increased mineralization.

Scintigraphy may also be performed to finetune diagnosis made by MRI and CT. The distal tarsus/proximal metatarsus region in the aging sport horse continues to be a challenging region with often pathological changes present in both on cross-sectional imaging. Increased uptake in the plantar proximal metatarsal region at the origin of the suspensory ligament or increased uptake related to the distal tarsal joints can be appreciated on bone scan and used for further management (Figure 4). Other examples in the distal limb where, in one of the authors’ experience, scintigraphy commonly assists in diagnosis making are the palmar/plantar processes of the distal phalanx, the navicular bone, and the sagittal groove of the proximal phalanx.

### 3.3. Scintigraphy and Longitudinal Follow-Up

In racehorses, scintigraphy is particularly helpful to follow-up activity of stress remodeling lesions in order to guide training intensity. In sport horses, follow-up is indicated for acute focal intense uptake, often of traumatic origin, in order to define further rehabilitation and intensification of work. However, regions of increased uptake in sport horses may remain present throughout the career of the horse, given the degenerative nature of many conditions. For this reason, longitudinal monitoring of scintigraphy findings with repetitive scans might be of limited clinical value in sport horses. Previously, a closer correlation between scintigraphy and current clinical presentation was expected, but the authors believe bone scan findings may also represent subclinical conditions indicative of clinical lesions a horse may be susceptible to develop.

## 4. Indications of Positron Emission Tomography

PET is the most recent modality to have become available for equine imaging. PET combines the functional aspect of nuclear medicine with the improved anatomical localization inherent to the cross-sectional imaging techniques. In brief, PET is to scintigraphy, what CT is to radiography. PET also presents the advantage of higher spatial resolution when compared with scintigraphy. Another particularity of PET is the availability of several radiotracers. With the availability of both bone (18F-NaF) and soft tissue (18F-FDG) tracers, a range of indications are available. As PET use in horses is relatively new, the most pertinent indications are still to be fully explored.

### 4.1. 18F-NaF PET as 3D Bone Scans of the Distal Equine Limb

The first large clinical 18F-NaF PET study focused on the racehorse fetlock [25]. This study demonstrated higher interobserver agreement for PET than for scintigraphy and PET detected more lesions than scintigraphy. This was particularly true for the proximal sesamoid bones, where PET detected abnormalities in 22.2% of the fetlocks when scintigraphy recognized abnormalities in only 6.9%, and also for the subchondral bone of the proximal phalanx [25]. PET also allowed better characterization of the site of uptake. Several different locations of uptake have been identified in the racehorse sesamoid bones, with the dorsal subchondral bone recognized as the most common [25].

In the sport horse fetlock, PET excels at identifying subchondral remodeling, in particular at the medial aspect and groove of the proximal phalanx and dorsodistal third metacarpal bone (Figure 5) [48]. These sites are known for the development of resorptive osseous lesions with poor prognosis once articular surface defects have developed, and PET provides an opportunity for early diagnosis.

18F-NaF PET of the foot is also particularly helpful with identification of navicular bone lesions, distal interphalangeal joint remodeling, and enthesopathies [7]. In addition to recognizing distal interphalangeal collateral ligament enthesopathies, PET demonstrated that remodeling can be identified at the attachment of the chondrosesamoidean ligament on the distal phalanx [49].

The design of the dedicated equine PET scanner allows for imaging as far proximal as the carpus and the tarsus [6]. For these two sites, PET demonstrates obvious advantages over scintigraphy with better definition of anatomical location of abnormalities. In racehorses, the carpus is commonly imaged with PET. Subchondral remodeling of the dorsoproximal aspect of the third carpal bone is common in racehorses and PET allows to characterize the extent and severity of the injury. Unlike scintigraphy, PET is able to distinguish uptake involving the distal aspect of the radial carpal bone, in addition to uptake of the third carpal bone. Tarsal PET imaging indications are common in sport horses, in particular for assessment of activity of distal tarsal joints degenerative changes (Figure 6), but PET also allows recognition of subchondral injuries of the distal tibia [50]. Another important role of PET in the distal tarsal and proximal metatarsal region is to localize the site of active lesion between distal tarsal joint or proximal suspensory, as PET is excellent at detecting active proximal suspensory enthesopathy (Figure 6).

### 4.2. 18F-NaF PET Combined with CT or MRI for Optimal Bone Imaging

PET provides functional information but lacks structural information. Combining PET with CT or MRI provides optimal assessment of the imaged area, benefiting from the advantages of functional and structural imaging. The value of adding PET to CT or MRI is twofold: (1) it allows the detection of early lesions, prior to the development of structural changes, and (2) it differentiates between active and inactive lesions in presence of structural changes [7]. An example for early lesion detection would be increased uptake at the flexor surface at the navicular bone when the contour and signal intensity remained within normal limits on CT and MRI. Distinguishing between active and inactive lesions can be appreciated at sites of enthesophyte formation: increased uptake associated with osteophytes suggests an active process, whereas uptake similar to background indicates a quiescent process [7].

Based on these principles, there are two main reasons to perform a PET after a CT or an MRI: (1) the structural imaging modalities did not find an explanation for the source of the lameness, and (2) abnormalities have been identified, but knowing the physiological status of the lesions (active versus inactive) is important to establish a treatment and rehabilitation plan. This can be particularly true when multiple lesions are identified, to decide which is the main contributor to the lameness.

PET combined with CT or MRI is particularly helpful for assessment of compact subchondral bone. Changes in the compact part of the subchondral bone can only be appreciated on CT or MRI once sufficient demineralization has occurred, so that a change in contour or signal intensity can be appreciated. MRI has the ability to detect early trabecular subchondral bone changes with the detection of fluid signal, but changes in the dense compact bone typically are not recognized until demineralization happen allowing for accumulation of fluid. Typically molecular changes occur before structural changes and PET allows early detection of compact subchondral bone remodeling (Figure 7). On the other hand, CT might show structural changes in the subchondral bone that are not currently active. PET in this regards provides additional information on the clinical significance of lesions.

PET is also very pertinent for the assessment of enthesopathies. As discussed for subchondral bone assessment, PET can recognize subtle enthesopathy, but another situation is assessment of an enthesis with irregular contour, as can commonly be the case with the suspensory origin. Irregular contour due the development of enthesophytes or focal areas of osseous resorption may remain after resolution of the activity of lesions making it challenging for clinical interpretation. This also applies to the collateral ligaments of the distal interphalangeal joint, as for example, to establish the clinical significance of changes in the contour of the fossa of attachment on the distal phalanx identified on CT or MRI (Figure 8).

### 4.3. 18F-FDG PET Combined with CT or MRI for Staging of Soft Tissue Lesions

Although the use of 18F-FDG PET remains limited in human medicine for specific musculoskeletal applications, a few studies suggest the value of this technique for detection of tendinopathy and monitoring of tendon healing [23,51]. Increased 18F-FDG uptake associated with equine deep digital flexore tendon (DDFT) lesion was reported in the original equine PET exploratory study [5]. A clinical study in eight horses with foot lameness compared PET findings with CT and MRI and confirmed the ability of PET to detect DDFT lesions [52]. Fewer lesions were detected with 18F-FDG PET than with CT and MRI, as only DDFT lesions with increased glucose metabolism can be recognized with PET. This limitation is important to recognize as chronic inactive lesions may still have clinical relevance; however, the knowledge of the metabolic activity is potentially pertinent to clinical management (Figure 9). 18F-FDG PET/CT has also been used in a small group of horses with suspected proximal suspensory desmitis [53,54]. Increased 18F-FDG uptake helps confirm activity of the desmitis when the US and CT changes are equivocal in chronic cases.

18F-FDG PET also has potential indications in the assessment of laminitis. Increased FDG uptake in the dorsal hoof wall but also decreased uptake in the coronary band were recognized in a population of horses with severe laminitis [55]. In chronic laminitic cases, PET can be helpful assessing the stage of the disease and the response to treatment.

### 4.4. PET for Longitudinal Monitoring of Lesions

Based on the functional nature of PET assessment, PET is particularly suitable for following lesions, both for assessing healing and monitoring recurrence. Some lesions will result in permanent structural changes making challenging assessment of healing over time with CT. Adding the functional component of PET is particularly helpful in such cases, for example, with suspensory or collateral ligament enthesopathies (Figure 8). Quantification of PET uptake can be performed using standardized uptake values (SUV) allowing more objective comparison between different scans. The SUV is a measure of the amount of radioactivity per area, taking into consideration the administered dose, the time between injection and image acquisition for decay correction, and the weight of the patient [56]. Maximal SUV (SUVmax) is the most commonly used value for assessment of lesions. Normal equine bone typically has an SUVmax for 18F-NaF between 3 and 5, whereas SUVmax of lesions can be as high as 60. In the soft tissues, the SUV for 18F-FDG remain much lower, due to a larger volume of distribution of the radiotracer, with normal tendon having an SUVmax of 1 or lower and lesions SUVmax typically ranging around 3 to 5. An association was found between 18F-NaF SUVmax and presence of fetlock pain, based on diagnostic analgesia, in a population of sport horses [48].

In racehorse fetlocks, the SUVmax of lesions at the time of diagnosis can be used to guide the duration of lay-up for resolution of increased uptake in lesions. In a longitudinal study in racehorses, lesions with SUVmax higher than three times the SUVmax of bone background were more likely to require more than 12 weeks off training to resolve [57]. 18F-NaF PET has also been used to monitor fetlock lesion recurrence in racehorses in training. Horses were scanned when returning to training after lay-up and 1, 2, 4, and 6 months later. Most horses did not show significant abnormalities at 1 and 2 months but abnormal uptake reoccurred in about half of the horses after 4 months. Abnormal uptake was more likely to develop at the site of the original lesion, that had been responsible for the lay-up, than in a different location [58].

## 5. Conclusions

The roles of nuclear medicine in equine lameness diagnosis have evolved since the introduction of scintigraphy in the late 1970s. Some of the original indications of equine scintigraphy are now outdated and have been replaced by other imaging techniques. Scintigraphy, however, retains a pertinent role due to some unique characteristics, such as the ability to image larger body parts and perform whole body imaging. The recent addition of PET opens a range of new applications in nuclear medicine. PET addresses some of the limitations of scintigraphy and also provides a useful addition to CT and MRI with identification of early changes and better staging of lesions. Both techniques require the use of radiopharmaceuticals leading to the application of radiation safety rules, limiting the use of the techniques to larger referral hospitals. The limited availability and high cost of the PET radiotracers in certain countries have for the moment remained an obstacle to development of equine PET outside of the USA, but these limitations will hopefully be overcome with the growth of PET use in human medicine.

## Figures and Tables

**Figure 1 animals-14-02499-f001:**
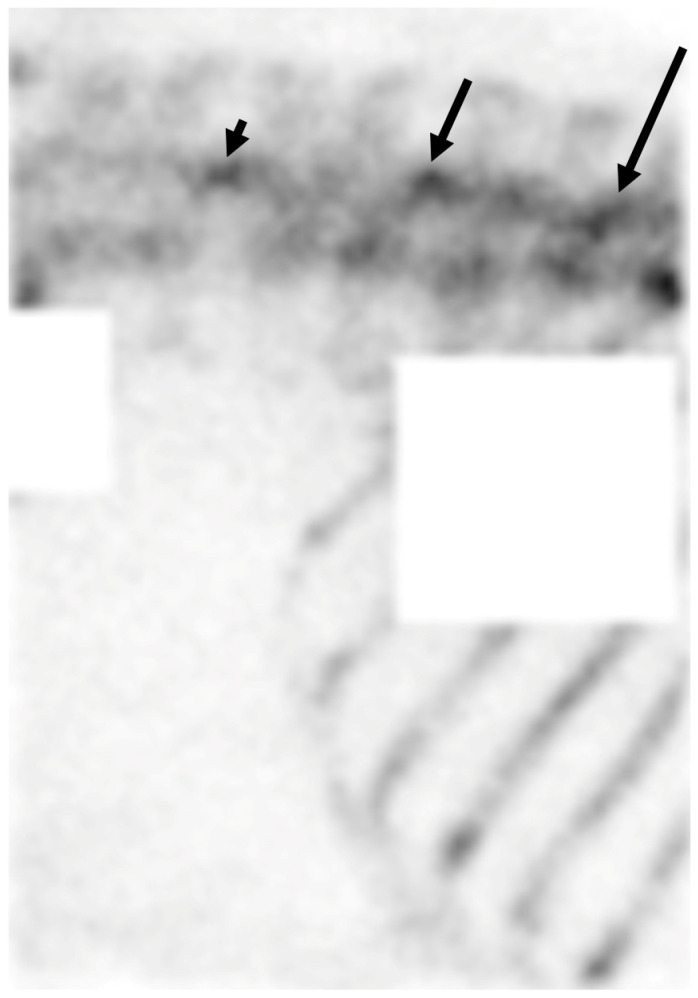
Scintigraphic image of the lumbar spine. Lateral oblique bone phase image of a 10-year-old dressage horse. There is focal increased uptake over the articular processes joint region between the 18th thoracic and first lumbar vertebrae (long arrow), the second and third lumbar vertebrae (short arrow), and the fourth and fifth lumbar vertebrae (arrow head).

**Figure 2 animals-14-02499-f002:**
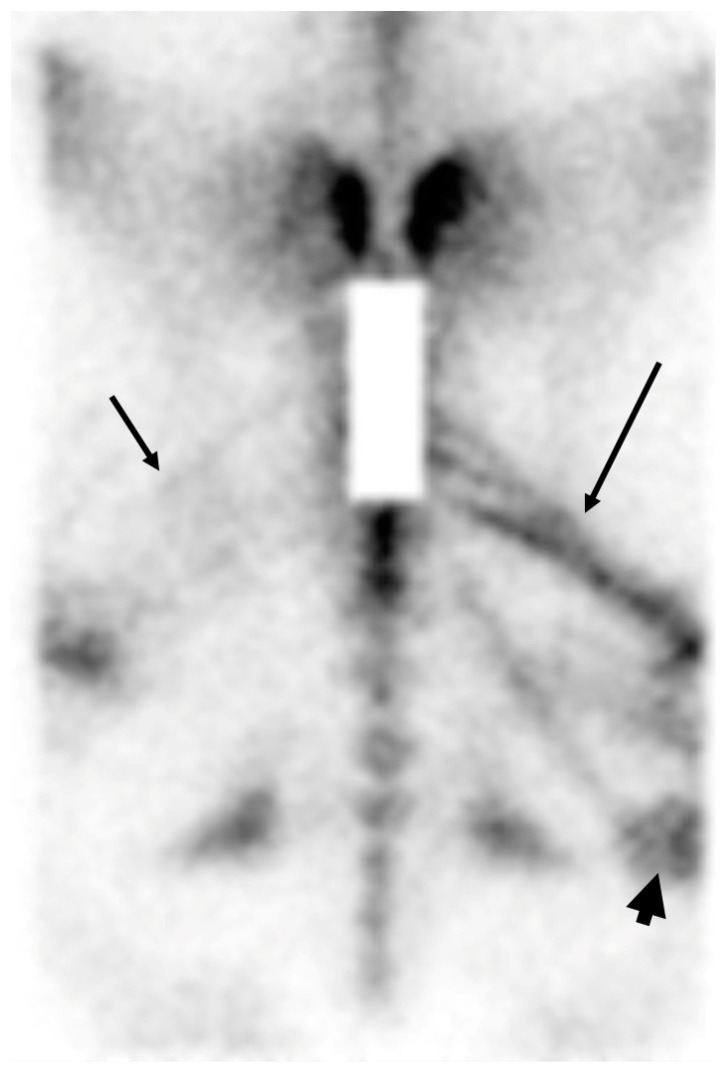
Scintigraphic muscular uptake. Dorsal bone phase image of the pelvis of an 8-year-old high-level show jumper, presented with reduced performance. There is linear increased radiopharmaceutical uptake in several muscle bundles of the gluteal and biceps femoral muscles: mild in the left gluteal muscles (short arrow) and severe in the right gluteal muscles (long arrow) and biceps femoral (arrowhead).

**Figure 3 animals-14-02499-f003:**
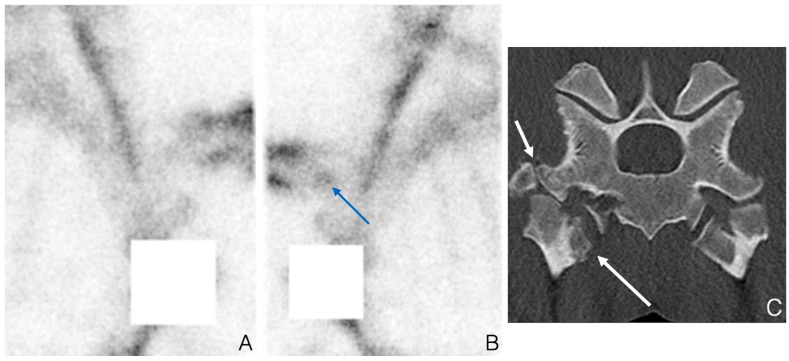
Scintigraphic and computed tomographic images of costovertebral remodeling. Lateral bone phase image of the right (**A**) and left (**B**) shoulder of a 9-year-old show jumper presented with left front lameness visible while being ridden. There is focal moderate increased uptake visible in the region of the first costovertebral joint (blue arrow). The transverse CT (**C**) image documents a large fragment of the left costotransverse joint (short arrow) and marked remodeling of the costotransverse and costovertebral joint of T1 (long arrow).

**Figure 4 animals-14-02499-f004:**
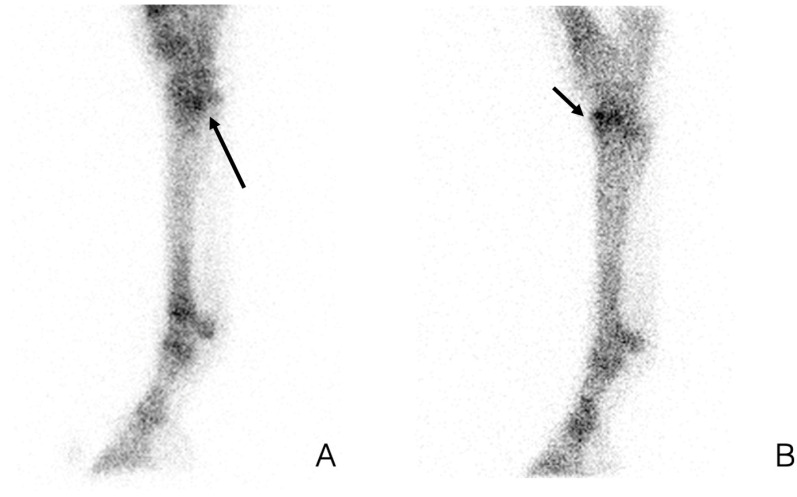
Tarsal scintigraphic images. Lateral scintigraphic bone phase images of the distal hind limbs from two different horses. Focal increased uptake at the plantar aspect of the proximal third metatarsal bone ((**A**), long arrow), several foci of increased uptake at the dorsal aspect of the central and third tarsal bone ((**B**), short arrow).

**Figure 5 animals-14-02499-f005:**
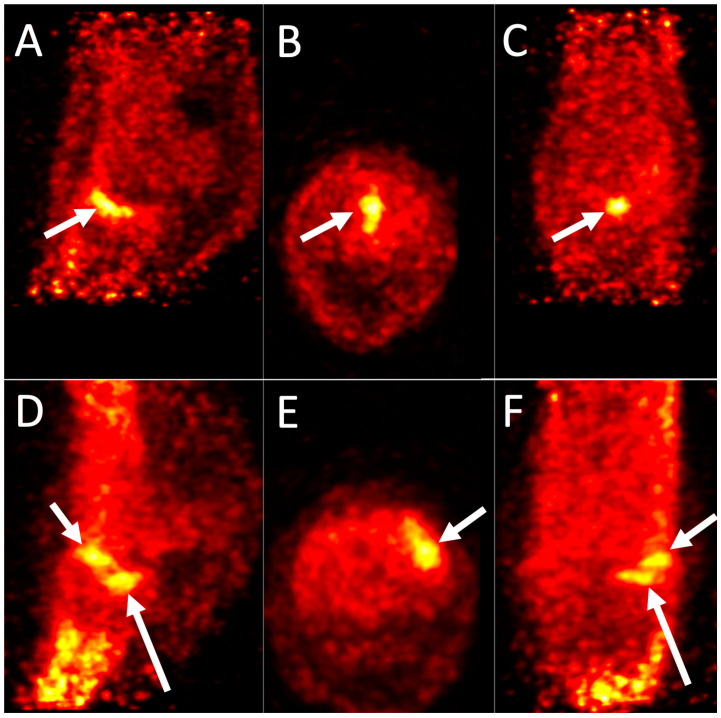
18F-NaF PET images of a sport horse fetlock. Three-dimensional-multiplanar reformat 18F-NaF PET images of the front fetlocks of two different horses. ((**A**,**D**): sagittal images, (**B**,**E**): transverse images, (**C**,**F**): dorsal images). Scans were performed on standing horses with a 4 min acquisition time. Top row: Left front fetlock of a 12-year-old Warmblood jumper with lameness resolved with an intra-articular metacarpophalangeal diagnostic analgesia. Medial is to the left. There is marked uptake in the subchondral bone of the groove of the proximal phalanx with a dorsopalmar orientation (Arrows). Bottom row: Right front fetlock of a 7-year-old Warmblood jumper with lameness resolved with an intra-articular metacarpophalangeal diagnostic analgesia. Lateral is to the left. There is marked uptake in the subchondral bone of the dorsomedial aspect of the metacarpal condyle (short arrow) and the medial fovea of the proximal phalanx (long arrow).

**Figure 6 animals-14-02499-f006:**
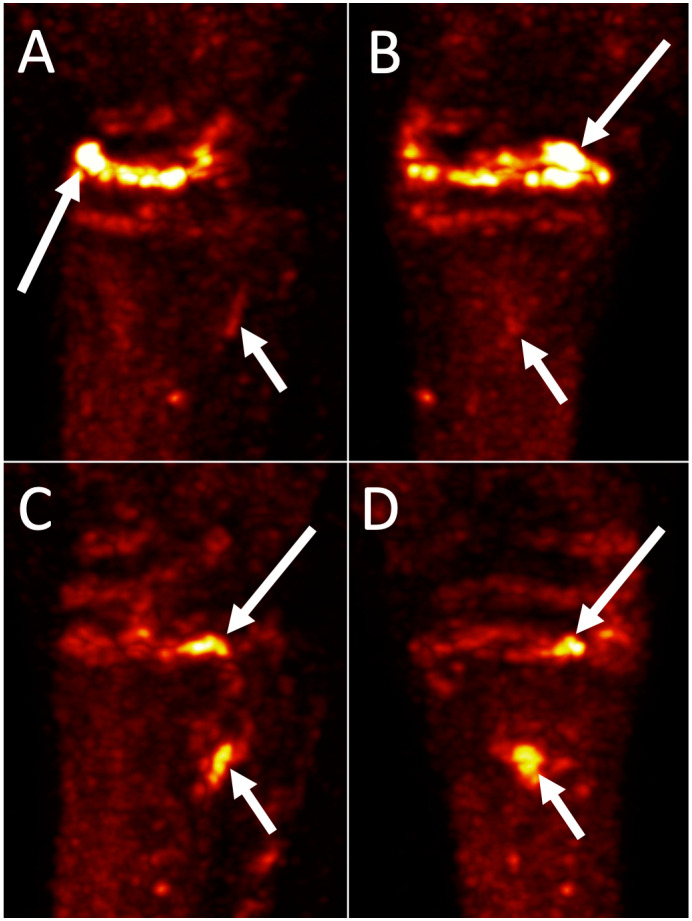
18F-NaF PET images of Quarter Horse Tarsi. Three-dimensional Maximal Intensity Projections ((**A**,**C**): lateral, (**B**,**D**): dorsal) of both tarsi of a 9-year-old Quarter Horse gelding with bilateral hindlimb lameness improved with perineural diagnostic analgesia of the deep branch of the lateral plantar nerve. Top row: Left tarsus, medial is to the left. There is marked increased 18F-NaF uptake through the distal intertarsal joint, worse dorsally and laterally (long arrow). There is minimal uptake at the plantar aspect of the third metatarsal bone at the origin of the suspensory ligament (short arrow). Bottom row: right tarsus, lateral is to the left. There is marked focal increased 18F-NaF uptake at the plantar medial aspect of the tarsometatarsal joint (long arrow). There is marked uptake at the plantar aspect of the third metatarsal bone at the origin of the suspensory ligament (short arrow).

**Figure 7 animals-14-02499-f007:**
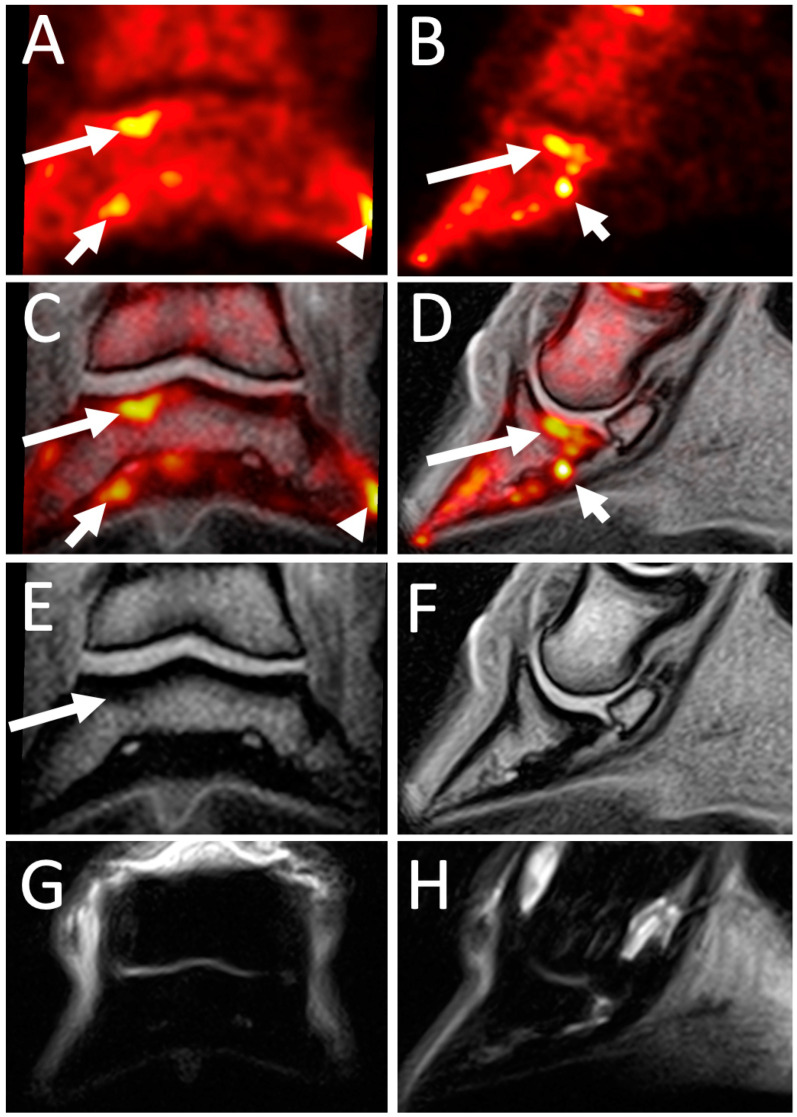
18F-NaF PET and MRI of a distal phalanx subchondral lesion. 18F-NaF PET MPR (**A**,**B**), fused PET / MRI MPR 3D Isotropic T1-weighted gradient echo (**C**,**D**), MRI 3D Isotropic T1-weighted gradient echo (**E**,**F**) and MRI STIR images (**G**,**H**) of the right front foot a 9-year-old Warmblood mare with a 3-week history of an acute 3/5 RF lameness that resolved following a right front palmar digital nerve block. The left column (**A**,**C**,**E**,**G**) are dorsal images with lateral aspect to the left of the image. The right column are sagittal images through the lateral aspect of the distal phalanx through the site of PET uptake. There is focal moderate increased 18F-NaF PET uptake within the compact subchondral bone plate at the lateral aspect of the distal phalanx (long arrow). There is mild focal non-specific increased thickness of the compact subchondral bone plate of the distal phalanx at the same site (long arrow) but no abnormality can be observed on the STIR images. Note that there is also mild to moderate increased uptake of the flexor surface of the distal phalanx (short arrow) and focal solar margin uptake medially (arrowhead).

**Figure 8 animals-14-02499-f008:**
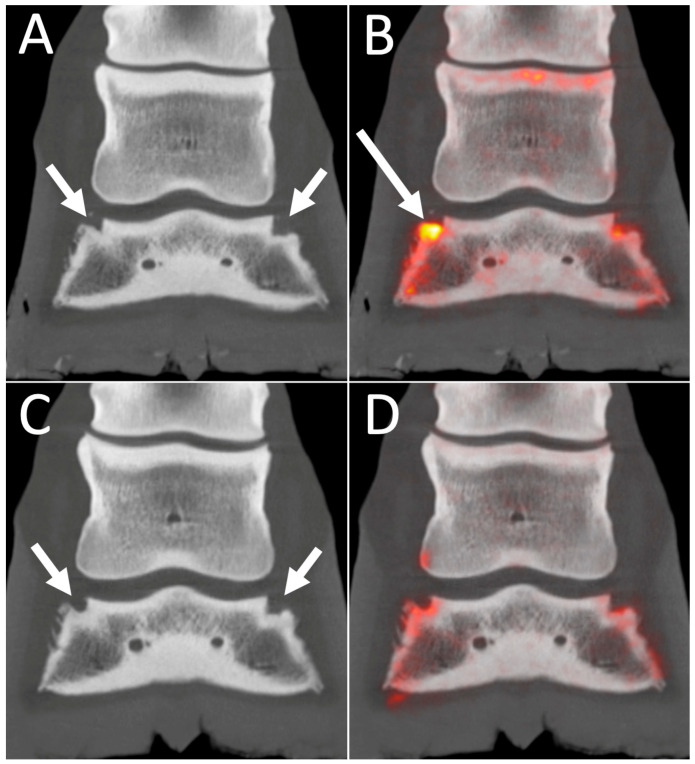
18F-NaF PET/CT images of collateral ligament enthesopathy. Dorsal plane CT (**A**,**C**) and fused 18F-NaF PET/CT (**B**,**D**) images of the right front foot of a 16-year-old Thoroughbred jumper with chronic front limb lameness and previous diagnosis of biaxial collateral ligament desmitis on MRI. Lateral is to the left. The top row images are the original scan and the bottom row a recheck scan 6 months later after resolution of the right front lameness. Top row: On the original images, there is evidence of biaxial remodeling of the distal phalanx at the sites of attachment of the collateral ligaments (short arrows) and marked focal uptake associated with the fossa of attachment of the lateral collateral ligament on the distal phalanx (Long arrow). Bottom row: On the recheck images, there is persistent evidence of biaxial remodeling of the distal phalanx at the sites of attachment of the collateral ligaments (short arrows); however, the 18F-NaF uptake has resolved providing good correlation with the clinical signs.

**Figure 9 animals-14-02499-f009:**
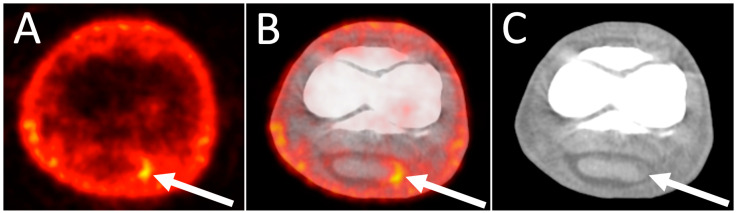
18F-FDG PET/CT images of tendon lesion. Transverse 18F-FDG PET (**A**), fused PET/CT (**B**), and CT (**C**) images though the right front pastern, at the level of the proximal interphalangeal joint, of a 15-year-old Warmblood jumper with right front lameness. Lateral is to the left. There is focal marked increased uptake at the abaxial aspect of the medial lobe of the deep digital flexor tendon (arrow). A hypoattenuating area is identified on CT in the same location (arrow).

## Data Availability

Not applicable.

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
