# Peer review of "Equine Nuclear Medicine in 2024: Use and Value of Scintigraphy and PET in Equine Lameness Diagnosis"

_animals, 2024, doi:10.3390/ani14172499_

Round 1

Reviewer 1 Report

Comments and Suggestions for Authors

Dear Authors,

I am reviewing your review article entitled “Equine Nuclear Medicine in 2024: Use and value of scintigraphy and PET in equine lameness diagnosis. I would thank the Authors for this interesting review; however, I have some suggestions to expand a little bit the manuscript.

Many sentences in the introduction section need at least a reference. In addition, I suggest to improve the initial part on the physic and principles of the bone scintigraphic images. A brief description of the three phases of acquisition, indications and pro and cons of them would be useful to be included for the review purpose.

Moreover, I suggest to expand the technical principles of the PET, which is not well known by all possible readers. This is partially included in lines 182-187. I suggest to move them in the introduction section. In the paragraph about combination of PET/MRI for bone imaging, there is poor connection with the clinical aspect or the clinical significance of the lesions. I mean what the PET add further to the combination of clinical presentation and MRI for example? Also MRI has some sequences that help in the evaluation of active or inactive lesion. Is the PET better? If yes, this should be stated clearer. In addition, especially in sports horses, some of these horses are able to perform with a correct management of some active lesions. In this scenario, the clinical presentation is important. How the PET help more than CT or MRI or scintigraphy in the authors opinion?  

Finally, I personally think that a section about limitations should be included.

Abstract

Fine

Introduction

Line 24-25: I suggest to specify in this sentence that these radiopharmaceutical agents link to hydroxyapatite crystals in order to allow evaluation of the bone activity. In addition, as this is a review, I suggest to describe that the images are proportional to the osteoblastic activity.

Line 31-34: authors reports that scintigraphy is still the “ONLY” imaging technique …. and remains the reference imaging tool for some…. pelvis and back imaging in sport horses. I suggest to rephrase as there is evidence in the literature that scintigraphy has some limitation for the evaluation of the back in sports horses (Quiney et al. 2018 Vet rad Ultrasound; Dyson 2014 Semin Nucl Med) and the advance in the use of US technique has demonstrated that scintigraphy is not the “ONLY” technique. Please expand.

Line 38: general anesthesia

Please expand the technical characteristics of PET.

Classic indications for bone scintigraphy

Line 49-50, 55-65: the first indication was right in the past. Now the advent of CT and MRI allow to use the scintigraphy less for this indication. This should be specified and clearly indicated. For this indication, scintigraphy has been surpassed.

Line 81-82: not sure proximal suspensory desmopathy should be included in “when lameness cannot be localized with diagnostic analgesia”. If clinically related to lameness, this disease should respond totally or partially to diagnostic analgesia available for the distal limb.

Line 131: I suggest to include ultrasound before muscle biopsy.

Line 166: sports horses (check the entire manuscript).

Indications of PET

Lines 182-187: Move these sentences in the introduction section and include here only the indications (lines 186-189).

Line 191: fetlock.[25] This…. Remove the dot after [25]

Line 199: sports horse; same in line 225

Line 201: third metacarpal bone (Figure 5). [26]

Line 202: “poor prognosis”. Do authors have a reference for this sentence. Or is it a personal experience? If so, specify. I do not have the same experience about “poor” prognosis if we consider the wide range of SCB lesions that can develop on this part of P1 and distal MC3, first of all the anatomical localization (more axially located or more abaxially located).

Line 228: Figure 6

Lines 245-246: I think that the characteristics of the clinical presentation is of great value, not only the diagnostic imaging findings. The same for the sentences of lines 257-262.

Line 248: MRI do not relay on demineralization or density. Please check this sentence. An example as for the CT would be useful for the MRI (lines 249-253).

Lines 288-293: so in this study in lame horses with DDFT tendinopathy, PET was able to exclude the DDFT as the source of lameness. What about the reason for foot lameness? This is a review, should be clarified.

Line 294-297: would the use of MRI have given the same result in your opinion? Same for lines 309-312. I mean: if a clinic does not have the MRI but only CT, add PET exam is a good option. Would be the same if the clinic has the MRI? Expand.

Line 313: enthesopathies (Figure 7).

Line 315: I am not able to find the SUV acronym per estenso. Add.

Figure legends

Figure 1. Scintigraphic image of the lumbar spine. Line 117: over the articular processes joint region….

Figure 2. Scintigraphic muscular uptake. Add the dot. Line 135: there is Line 137 and gluteofemoral muscle (Nomina anatomica)

Figure 3. Scintigraphic and computed tomographic images. Line 151: 9-year-old. Line 152: there is … The cross sectional CT image (C)…

Figure 4. Tarsal scintigraphic image. line 178: (A), several

Figure 5. Line 205. horse fetlock. Line 207: images). Line 209: metacarpophalangeal diagnostic analgesia. Same in line 212. Line 213: medial part

Figure 6: Line 264: Tarsi. line 266: perineural diagnostic anesthesia… lateral plantar nerve

Figure 7: enthesopathy.

Figure 8: tendon lesion.

References.

reference 7 need to follow the references style.

Author Response

Reviewer 1:

I am reviewing your review article entitled “Equine Nuclear Medicine in 2024: Use and value of scintigraphy and PET in equine lameness diagnosis. I would thank the Authors for this interesting review; however, I have some suggestions to expand a little bit the manuscript.

Many sentences in the introduction section need at least a reference. In addition, I suggest to improve the initial part on the physic and principles of the bone scintigraphic images. A brief description of the three phases of acquisition, indications and pro and cons of them would be useful to be included for the review purpose.

We added some technical information, however we see the focus of this manuscripts more on the clinical indications so we kept the technical aspect fairly short.

Moreover, I suggest to expand the technical principles of the PET, which is not well known by all possible readers. This is partially included in lines 182-187. I suggest to move them in the introduction section.

Same comment as for scinti, we added a small amount of technical info. We opted to have it at the beginning of the PET section to keep the intro fairly succinct.

 In the paragraph about combination of PET/MRI for bone imaging, there is poor connection with the clinical aspect or the clinical significance of the lesions. I mean what the PET add further to the combination of clinical presentation and MRI for example? Also MRI has some sequences that help in the evaluation of active or inactive lesion. Is the PET better? If yes, this should be stated clearer. In addition, especially in sports horses, some of these horses are able to perform with a correct management of some active lesions. In this scenario, the clinical presentation is important. How the PET help more than CT or MRI or scintigraphy in the authors opinion?  

We developed this to try to clarify. We added a figure.

Finally, I personally think that a section about limitations should be included.

Rather than writing a specific limitation section, we mentioned through the texts the limitations of NM and included some of these in the conclusion.

Abstract

Fine

Introduction

Line 24-25: I suggest to specify in this sentence that these radiopharmaceutical agents link to hydroxyapatite crystals in order to allow evaluation of the bone activity. In addition, as this is a review, I suggest to describe that the images are proportional to the osteoblastic activity.

We wish to keep this introduction paragraph relatively concised and focused on the history of scintigraphy and opted to not add here technical details. We added more technical information at the beginning of the scintigraphic part of the manuscript.

Line 31-34: authors reports that scintigraphy is still the “ONLY” imaging technique …. and remains the reference imaging tool for some…. pelvis and back imaging in sport horses. I suggest to rephrase as there is evidence in the literature that scintigraphy has some limitation for the evaluation of the back in sports horses (Quiney et al. 2018 Vet rad Ultrasound; Dyson 2014 Semin Nucl Med) and the advance in the use of US technique has demonstrated that scintigraphy is not the “ONLY” technique. Please expand.

“Only”referred to whole body imaging. We have rephrased to avoid this confusion.

Line 38: general anesthesia

Please expand the technical characteristics of PET.

Similar to the scintigraphy comment we opt to not expend the introduction with technical information. We added some of these later.

Classic indications for bone scintigraphy

Line 49-50, 55-65: the first indication was right in the past. Now the advent of CT and MRI allow to use the scintigraphy less for this indication. This should be specified and clearly indicated. For this indication, scintigraphy has been surpassed.

This is exactly the point we wanted to make. We replaced “classic” with “original” and developed a bit the text to clarify this further.

Line 81-82: not sure proximal suspensory desmopathy should be included in “when lameness cannot be localized with diagnostic analgesia”. If clinically related to lameness, this disease should respond totally or partially to diagnostic analgesia available for the distal limb.

Fair point, the reason it is in this paragraph is that it is further explanation on the prior sentence that comments on the overall sensitivity and specificity of scinti in sport horses.

Line 131: I suggest to include ultrasound before muscle biopsy.

Added

Line 166: sports horses (check the entire manuscript).

Unsure about this comment?

Indications of PET

Lines 182-187: Move these sentences in the introduction section and include here only the indications (lines 186-189).

We wish to keep the overall intro succinct and prefer have a PET intro here with some of the requested technical information.

Line 191: fetlock.[25] This…. Remove the dot after [25]

Done

Line 199: sports horse; same in line 225

I believe “sport horses” is the appropriate spelling

Line 201: third metacarpal bone (Figure 5). [26]

Done

Line 202: “poor prognosis”. Do authors have a reference for this sentence. Or is it a personal experience? If so, specify. I do not have the same experience about “poor” prognosis if we consider the wide range of SCB lesions that can develop on this part of P1 and distal MC3, first of all the anatomical localization (more axially located or more abaxially located).

We were not able to find a specific reference but have specified our impressions.

Line 228: Figure 6

Done

Lines 245-246: I think that the characteristics of the clinical presentation is of great value, not only the diagnostic imaging findings. The same for the sentences of lines 257-262.

We agree that clinical presentation is of great value and did not say otherwise.

Line 248: MRI do not relay on demineralization or density. Please check this sentence.

I agree in general but the comment here is regarding the compact subchondral bone where we believe some demineralization is needed to appreciate changes in MRI. We developed to clarify.

 An example as for the CT would be useful for the MRI (lines 249-253).

We have rephrased and added a PET MRI figure illustrating our point regarding compact subchondral bone injuries.

Lines 288-293: so in this study in lame horses with DDFT tendinopathy, PET was able to exclude the DDFT as the source of lameness. What about the reason for foot lameness? This is a review, should be clarified.

Rephrased.

Line 294-297: would the use of MRI have given the same result in your opinion? Same for lines 309-312. I mean: if a clinic does not have the MRI but only CT, add PET exam is a good option. Would be the same if the clinic has the MRI? Expand.

Clarified with the addition of a figure

Line 313: enthesopathies (Figure 7).

Done

Line 315: I am not able to find the SUV acronym per estenso. Add.

Done

Figure legends

Figure 1. Scintigraphic image of the lumbar spine. Line 117: over the articular processes joint region….

Figure 2. Scintigraphic muscular uptake. Add the dot. Line 135: there is Line 137 and gluteofemoral muscle (Nomina anatomica)

Figure 3. Scintigraphic and computed tomographic images. Line 151: 9-year-old. Line 152: there is … The cross sectional CT image (C)…

Figure 4. Tarsal scintigraphic image. line 178: (A), several

Figure 5. Line 205. horse fetlock. Line 207: images). Line 209: metacarpophalangeal diagnostic analgesia. Same in line 212. Line 213: medial part

Figure 6: Line 264: Tarsi. line 266: perineural diagnostic anesthesia… lateral plantar nerve

Figure 7: enthesopathy.

Figure 8: tendon lesion.

The figure legends have been corrected.

References.

reference 7 need to follow the references style.

This is the format this journal used to have and appears this way in all downloadable citations tools. We have manually fixed it.

Reviewer 2 Report

Comments and Suggestions for Authors

Please see the attached comments, with many thanks

Comments on the Quality of English Language

Just a small number of grammatical errors noted. 

Author Response

This is a really nice review articular providing an overview of the role of two forms of nuclear scintigraphy available for equine patients. The examples provided are well chosen, clear and well annotated.
Thank you!

I have spotted a couple of minor grammatical errors; Line 94 suggest exchange “to” for the word ”for” Lines 127-128 suggest rewording “to identified” Line 233 change “lesion” to “lesions”
Corrected

Line 243 point 2. I would argue that whilst CT cannot easily distinguish between chronic or acute conditions, MRI is able to provide evidence that a lesion is active versus chronic, by means of assessment of the presence of any degree of abnormal increased fluid component within tissues. For example, in figure 7, I would anticipate that images from STIR pulse sequences would show abnormal increased fluid within the lateral collateral fossa and adjacent body of P3. Can the authors comment please?
We have developed this explanation and illustrated with an additional figure. MRI is limited in demonstrating active changes in the compact bone, either in subchondral bone or at ligament attachments. If the changes extend to the trabecular bone, typically hyperintense signal can be appreciated on STIR but we have CL DIPJ cases that have increased uptake on PET in the fossa without appreciated STIR hyperintensity.

Reference 7. – unclear why this is in capitals?
Fixed, this is the format VRU used to have.

Otherwise I have no concerns.
Thank you.

Reviewer 3 Report

Comments and Suggestions for Authors

This article addressed the use of scintigraphy and PET in sports horses and racehorses predominantly (seems to have forgotten that a very large proportion of horses are used for leisure purposes). The authors do  not set out their aims and objectives. This is not a systematically performed review, performed as advised by PRISMA guidelines. It appears more to be a personal opinion piece which lacks depth & critical appraisal of the available literature.  In my opinion this paper requires radical rewriting. It is not acceptable in its current form. The PET section is considerably better than the section on scintigraphy but in my opinion still needs major improvements.

See The PRISMA 2020 statement: an updated guideline for reporting systematic reviews

BMJ 2021; 372 doi: https://doi.org/10.1136/bmj.n71 (Published 29 March 2021)

Cite this as: BMJ 2021;372:n71

There are some good figures but some of the legends need improving.

Lines 14-15

This manuscript will cover current use and values of these two modalities in equine lameness diagnosis. ‘

The Abstract should be a summary of the content.

Line 24 'Scintigraphy in horses is primarily used for bone imaging with Technetium-99m (99mTc) 24 coupled with bisphosphonates agent.'  Doesn't really make sense.

Scintigraphy is a method of bone imaging utilising a bone-seeking agent, a  bisphosphonate, labelled with technetium-99m (99mTc).

Line 30 'years' is redundant

Line 34 'back' is not a very scientific term - thoracolumbosacral and pelvic regions.

Line 47 laminae

Line 57 pertinent - useful might be a better word; pertinent is used far too many times throughout the section on scintigraphy.

Line 63 & elsewhere  uptake of what?

Lines 66 onwards This is not really a review.

What were the objectives of this paper? It is not a systematic review.

The scintigraphy section seems to be  a simplified author selected overview of a rather arbitrary selection of scintigraphy-related publications.

Line 73-75 But of what clinical relevance?

Consider whether IRU is synonymous with pain

There are sites at which focal intense IRU can be seen with no associated clinical signs  e.g., the dorsoproximal metaphyseal region of the proximal phalanges.

Line 84 see the variety of publications concerning scintigraphy and the  sacroiliac joints.

Line 93 What is the evidence for the statement '... & even regains interest in the evaluation of poor performance in sport horses.' ?

Line 94-96 Not based on 18! If you look at all the published material on scintigraphy of the SI joints it is clear that there is a huge overlap between what can be seen in normal horses and those with LSI joint region pain and a high proportion of false negative results.

Line 96 'pertinent' is an over-used word in the entire section on scintigraphy.

The section lacks balance and fails to adequately recognise the studies in which scintigraphy has provided potentially misleading results.

Line 104 onwards Doesn't critically evaluate the literature nor present the results of good studies.

Inadequate reference to related studies.

Line 110 Please read critically what you have written

'or shape or the enlargement ' - shape or enlargement of what?

Reference  is made repeatedly to 'uptake' without specifying uptake of what.

Line 112 Articular process joints (APJs)

Line 113 How can 'Left-right asymmetry in uptake ' be a 'cause  of asymmetrical gait or work'?

Line 117 uptake of what? Radiopharmaceutical uptake is not into joints but in the articular processes

Line 120 'Imaging of ribs remains challenging radiographically'   It is quite easy to do - the question is when do you do it - what prompts you to do it?

Line 121 Sensitivity was not addressed.

The horses underwent other diagnostic imaging based on identification of IRU.

Line 123 Synostoses of the first 2 ribs are easily seen radiographically.

Line 124 Easy to acquire  radiographs of the shoulder and coxofemoral joint  & perform ultrasonography.

Line 127 onwards  This section on muscle highlights the absence of a systematic review of recent literature

e.g.,Equine Vet Educ. 2023 DOI: 10.1111/eve.13920 Increased radiopharmaceutical uptake in skeletal muscle in 26 flat racing and endurance horses (2017–2021).

Line 131 But is this always of clinical ? - refer to other later studies. And absence of IRU does not preclude chronic RER.

Line 138  onwards    Where are the references to papers in which the combined use of MRI & scintigraphy has been described?

Line 146 What is meant by  'increased bone fluid like STIR signal'?

Figure 3 Provide orientation of image C. In legend amend Y to year.

Line 157 ageing

What is a show horse?

Line 163 distal phalanx, not P3

proximal phalanx, not P1

No references - this seems like a personal opinion piece rather than a proper systematic review

Line 171 what is meant by 'disappointing' - of low clinical value?

Lines 172- 174 ‘Previously a closer correlation between scintigraphy and current clinical presentation was expected, but bone scan findings may also represent subclinical conditions  indicative of clinical lesions a horse may be susceptible to develop. ‘     -   what evidence do you have to support this?

Line 176 IRU does not equate with pain & lameness

Can have IRU in tarsal bones in horses performing normally

Murray R, Dyson S, Weekes J, Short C & Branch M.  Nuclear scintigraphic evaluation of the distal  tarsal region in normal horses. Vet. Radiol Ultrasound 2004, 45: 345 – 351

Line 179 What is meant by 'In the older active athlete scintigraphy can help to discriminate in pain.....'?

Scintigraphy can identify different regions of IRU which may be associated with pain and lameness.

Line 216 What is meant by 'navicular bone changes'?

Line 228 'PET is able to distinguish active from inactive distal tarsal osteoarthritis 'Where is the evidence?

Line 235 Ref 4 based on 3 horses

Line 236 contour and density - how does this relate to MRI?

Line 248 'Changes can only be appreciated on CT or MRI once sufficient demineralization has occurred, so ..'  But what about hyperintense signal on an MR image??  How can you use MRI to assess  demineralization?

Line 268 in the subchondral bone of the centrodistal joint

The radiopharmaceutical uptake is not into a joint

Line 271 - as above - in subchondral bone not the joint

Line 276 At what level in the CLs? What abnormalities were seen on the MR images?Line 287 Had, not showing

Line 292 But this tells us nothing about pain - & lameness – necessarily

Line 296 29 = Proceedings 2021 ACVS Surgery Summit. Cannot access this.

Wouldn't the AJVR article be better?- peer reviewed and accessible

Line 301 PET can be helpful assessing the stage of the disease and the response to treatment

How?? Give evidence; explain

Line 314 with using  - with or using - not both

Line 366 'Nuclear medicine regroups the oldest and newest advanced equine imaging techniques. '  This seems an odd sentence for the conclusions.

References: Inconsistency of reference formats – please amend

Comments on the Quality of English Language

see above

Author Response

This article addressed the use of scintigraphy and PET in sports horses and racehorses predominantly (seems to have forgotten that a very large proportion of horses are used for leisure purposes).
We used “sport horses” in the broad sense of the term to cover all non racehorse activities.

The authors do  not set out their aims and objectives.
These have now been clarified.

 This is not a systematically performed review, performed as advised by PRISMA guidelines. It appears more to be a personal opinion piece which lacks depth & critical appraisal of the available literature.  In my opinion this paper requires radical rewriting. It is not acceptable in its current form. The PET section is considerably better than the section on scintigraphy but in my opinion still needs major improvements.
This manuscript was indeed never intended as a systematically performed review. Several high quality review papers already exist about equine scintigraphy. The literature on equine PET remains limited, several current clinical applications have not yet been published.  This was an invited manuscript and based on the suggested title “Equine NM in 2024”, the authors, who are experienced imaging clinicians, understood they were asked to in part provide their opinion on the current roles of scintigraphy and PET. We believe the originality of this paper primarily resides indeed in the clinical expertise of the authors.
After reading your review, we contacted the editors who had invited us to write this paper and we suggested retracting our manuscript. The editors convinced us otherwise, comforting us in the way we had initially interpreted the request.
We thank you for your comments and have as much as possible addressed them, but again this is definitely not a systematic review.

See The PRISMA 2020 statement: an updated guideline for reporting systematic reviews

BMJ 2021; 372 doi: https://doi.org/10.1136/bmj.n71 (Published 29 March 2021)

Cite this as: BMJ 2021;372:n71

There are some good figures but some of the legends need improving.
Legends have been revised.

Lines 14-15 ‘This manuscript will cover current use and values of these two modalities in equine lameness diagnosis. ‘The Abstract should be a summary of the content.

 In a 100 word abstract, it would be difficult to summarize the content. We would consider it if a longer abstract is possible but the rest of the reviewer team seems satisfied with the current “abstract”.

Line 24 'Scintigraphy in horses is primarily used for bone imaging with Technetium-99m (99mTc) 24 coupled with bisphosphonates agent.'  Doesn't really make sense.

 Scintigraphy is a method of bone imaging utilising a bone-seeking agent, a  bisphosphonate, labelled with technetium-99m (99mTc).

We respectfully disagree but our statement “makes more sense” than the suggested statement. Scintigraphy has many applications outside of bone imaging in human and veterinary medicine.

Line 30 'years' is redundant

 Deleted

Line 34 'back' is not a very scientific term - thoracolumbosacral and pelvic regions.

 Replaced with vertebral column

Line 47 laminae

 Corrected

Line 57 pertinent - useful might be a better word; pertinent is used far too many times throughout the section on scintigraphy.

 Changed to useful

Line 63 & elsewhere  uptake of what?

 Increased radiopharmaceutical uptake. Some authors like to call “IRU”

Lines 66 onwards This is not really a review.

What were the objectives of this paper? It is not a systematic review.

The scintigraphy section seems to be  a simplified author selected overview of a rather arbitrary selection of scintigraphy-related publications.

 See reply to your general comments

Line 73-75 But of what clinical relevance? Consider whether IRU is synonymous with pain. There are sites at which focal intense IRU can be seen with no associated clinical signs  e.g., the dorsoproximal metaphyseal region of the proximal phalanges.

 We have revised to clarify this point.

Line 84 see the variety of publications concerning scintigraphy and the  sacroiliac joints.

 We added comment on false negative with reference to Barstow EVE 2015

Line 93 What is the evidence for the statement '... & even regains interest in the evaluation of poor performance in sport horses.' ?

 We removed this statement. It was an opinion of one of the authors based on his clinical practice experience.

Line 94-96 Not based on 18! If you look at all the published material on scintigraphy of the SI joints it is clear that there is a huge overlap between what can be seen in normal horses and those with LSI joint region pain and a high proportion of false negative results.

 We mentioned again the risk for false negative to clarify.

Line 96 'pertinent' is an over-used word in the entire section on scintigraphy.

 We rephrased with “useful”.

The section lacks balance and fails to adequately recognise the studies in which scintigraphy has provided potentially misleading results.

Line 104 onwards Doesn't critically evaluate the literature nor present the results of good studies.

Inadequate reference to related studies.

 We apologize if this paragraph isn’t on par with the standards and knowledge of the reviewer. An exhaustive review of the scintigraphic literature is beyond the scope of what we had agreed when asked to write this manuscript and we currently do not have the bandwidth to perform this task.

Line 110 Please read critically what you have written

'or shape or the enlargement ' - shape or enlargement of what?

Revised

Reference  is made repeatedly to 'uptake' without specifying uptake of what.

 Radiopharmaceutical uptake

Line 112 Articular process joints (APJs)

 Corrected

Line 113 How can 'Left-right asymmetry in uptake ' be a 'cause  of asymmetrical gait or work'?

 This is a clinical observation from one of the authors.

Line 117 uptake of what? Radiopharmaceutical uptake is not into joints but in the articular processes

 Corrected

Line 120 'Imaging of ribs remains challenging radiographically'   It is quite easy to do - the question is when do you do it - what prompts you to do it?

 Correct

Line 121 Sensitivity was not addressed.

The horses underwent other diagnostic imaging based on identification of IRU.

It mentions scintigraphy identifying all fracture sites.  

Line 123 Synostoses of the first 2 ribs are easily seen radiographically.

 Correct

Line 124 Easy to acquire  radiographs of the shoulder and coxofemoral joint  & perform ultrasonography.

 Revised

Line 127 onwards  This section on muscle highlights the absence of a systematic review of recent literature

e.g.,Equine Vet Educ. 2023 DOI: 10.1111/eve.13920 Increased radiopharmaceutical uptake in skeletal muscle in 26 flat racing and endurance horses (2017–2021).

  Reference added.

Line 131 But is this always of clinical ? - refer to other later studies. And absence of IRU does not preclude chronic RER.

Agree but no pretention to this being an exhaustive review.

Line 138  onwards    Where are the references to papers in which the combined use of MRI & scintigraphy has been described?

 This paragraph is indeed more the presentation of a clinical opinion and not a systematic review. There are indeed many publications comparing scintigraphy and MRI but this is not the scope of this paper to review them all.

Line 146 What is meant by  'increased bone fluid like STIR signal'?

 Revised

Figure 3 Provide orientation of image C. In legend amend Y to year.

  Done

Line 157 ageing

Corrected

What is a show horse?

 Revised

Line 163 distal phalanx, not P3

proximal phalanx, not P1

 Corrected

No references - this seems like a personal opinion piece rather than a proper systematic review

 Correct

Line 171 what is meant by 'disappointing' - of low clinical value?

 Revised

Lines 172- 174 ‘Previously a closer correlation between scintigraphy and current clinical presentation was expected, but bone scan findings may also represent subclinical conditions  indicative of clinical lesions a horse may be susceptible to develop. ‘     -   what evidence do you have to support this?

 Clarified this is an opinion

Line 176 IRU does not equate with pain & lameness

Can have IRU in tarsal bones in horses performing normally

Murray R, Dyson S, Weekes J, Short C & Branch M.  Nuclear scintigraphic evaluation of the distal  tarsal region in normal horses. Vet. Radiol Ultrasound 2004, 45: 345 – 351

Revised

Line 179 What is meant by 'In the older active athlete scintigraphy can help to discriminate in pain.....'?

Scintigraphy can identify different regions of IRU which may be associated with pain and lameness.

 This statement has been removed.

Line 216 What is meant by 'navicular bone changes'?

 Rephrased

Line 228 'PET is able to distinguish active from inactive distal tarsal osteoarthritis 'Where is the evidence?

 Rephrased

Line 235 Ref 4 based on 3 horses

 Correct

Line 236 contour and density - how does this relate to MRI?

 Corrected

Line 248 'Changes can only be appreciated on CT or MRI once sufficient demineralization has occurred, so ..'  But what about hyperintense signal on an MR image??  How can you use MRI to assess  demineralization?

 Revised for further clarification

Line 268 in the subchondral bone of the centrodistal joint

The radiopharmaceutical uptake is not into a joint

 Indeed the uptake is in the subchondral bone, but we did not write the uptake to be in the “joint space”. The subchondral bone is part of the joint.

Line 271 - as above - in subchondral bone not the joint

 See above

Line 276 At what level in the CLs? What abnormalities were seen on the MR images?

This was the history we had and unfortunately the MRI was not available to us.

Line 287 Had, not showing

 Corrected

Line 292 But this tells us nothing about pain - & lameness – necessarily

 Correct but this remains pertinent in case management

Line 296 29 = Proceedings 2021 ACVS Surgery Summit. Cannot access this.

Wouldn't the AJVR article be better?- peer reviewed and accessible

 Added

Line 301 PET can be helpful assessing the stage of the disease and the response to treatment

How?? Give evidence; explain

 Rephrased

Line 314 with using  - with or using - not both

 Corrected

Line 366 'Nuclear medicine regroups the oldest and newest advanced equine imaging techniques. '  This seems an odd sentence for the conclusions.

 Rephrased

References: Inconsistency of reference formats – please amend

Fixed

Round 2

Reviewer 3 Report

Comments and Suggestions for Authors

I have carefully re-read the entire manuscript and remain concerned about the scintigraphy section. Any published paper should provide accurate information.  It will after all potentially be cited in the future. A review should cite appropriate recent literature, critically discussed in the light of the authors’ experience,  even if the authors were invited to in part provide their opinion on the current roles of scintigraphy and PET.  There are studies that have critically looked at the usefulness of scintigraphy in various anatomical locations and this paper should include these either to support the authors’ experiences or the authors need to point out in what circumstances their experiences have been different.

The authors point out that in the distal aspect of  limbs other imaging techniques have largely superseded scintigraphy, but point out a few instances in which scintigraphy may be valuable – but these are largely not supported by references – and references are available and should be cited. Moreover, the authors should make it clear how scintigraphy can potentially provide useful information in those scenarios.

Likewise in the vertebral column and pelvic regions, and proximal aspects of the limbs there are references demonstrating the clinical usefulness of scintigraphy which have not been included and need to be.

Any discussion of the value of scintigraphy should make it clear that IRU is not synonymous with pain causing lameness or poor performance, and that IRU may persist long-term, so the presence of IRU must be interpreted with care. It may also be influenced by exercise history and work discipline.

I therefore consider that the scintigraphy section needs fairly major revision before this paper could be considered acceptable for publication.

Line 50 This would read better as 'In particular, the aim is to illustrate, based on the authors' clinical experience, ....'

I still feel that it is necessary to provide references to relevant literature  to support your clinical observations and to acknowledge where your clinical experience is at variance from the published literature.

Any published paper is liable to be quoted in the future and there is a danger of perpetuating information which is not based on properly conducted clinical observational studies, but just on remembered observations.

Subsection 2 Based on the way this is now titled 'original indications for scintigraphy' you need to clarify in what situations you no longer feel it is relevant - which you do for tarsal scintigraphy & a central tarsal bone fracture but not for fetlock imaging in a racehorse - the reader is left 'dangling' – not knowing what your experience is, nor what the literature tells us.

Lines 68-69 Reference needed

Line 81 So how are you suggesting that aortoiliacofemoral thrombosis  should be diagnosed now?

Line 82 'has been attempted'

This is potentially misleading - see Dyson S, Murray R, Schramme M & Branch M.  Lameness in 46 horses associated with deep digital  flexor tendonitis in the digit: diagnosis confirmed with magnetic resonance imaging. Equine vet. J. 2003, 35: 681 - 690

Lines 89 & elsewhere ‘pertinent’ is still repeated numerous times

Lines 93-5 More relevant references need to be given because this does not read like personal clinical experience based on facts. If you can provide clinical data that would be useful as well! – supportive of your statements.

Lines 101-3 So what does high specificity but low sensitivity actually imply with respect to usefulness?

Line 108 false negative results

Line 110 repetition

Line  117 ‘but under the right circumstances, acknowledging the risk for  false negatives, whole body scintigraphy remains a useful tool    

So what are the ‘right circumstances’ ?   You cite a 2007 reference - & Archer et al was a review of previous publications , before 2007 - we are now in 2024! – and other publications based on clinical evidence have superseded that!   And what about false positive results?

Line 123 'it appears that a combination of different modalities  provides the most comprehensive approach to complex lameness issues ' - but you have not really provided many examples - in the first paragraph you compare scintigraphy with radiography - this is not new!

Line 135 'Left-right asymmetry in uptake in the lumbar articular facets is an important parameter to consider as a cause or consequence of asymmetrical gait or work.'

Increased radiopharmaceutical uptake does not cause lameness - it may be associated with lameness.

Please be consistent in nomenclature – the correct term is articular process joints – and the IRU is in the bone – so in the articular processes.

Line 138 Imaging of ribs is not challenging – you concurred with this previously yet have not changed the text – this is annoying.

Line 142 repetition

Lines 146 - 150 but IRU in muscle may be an incidental finding and in some horses with clinically significant myopathy scintigraphy gives false negative results – the information delivered must be accurate.

Line 161 'Especially in sport horses a mixture of signal intensities can be seen in degenerative pathologies. ' - what is this supposed to mean?

Lines 168- 170 ‘Other examples  in the distal limb where scintigraphy commonly assists in diagnosis making are the palmar/plantar processes of the distal phalanx, the navicular bone and the sagittal groove of the proximal phalanx.’    How? References needed.  And when increased signal intensity in STIR images is the main finding from MRI scintigraphy is usually negative.

Line 176 ‘However regions of increased uptake in sport horses may remain present throughout the career of the horse, given the degenerative nature of many conditions.’  This is an odd statement.  IRU associated with an exostosis on McII for example may persist long-term – this is not a degenerative condition. IRU in a tuber ischium may be present long term – this is not a degenerative condition.

Line 222 ‘Another important role of PET in the distal tarsal and proximal metatarsal region is to distinguish the origin of pain in the area between distal tarsal joint or proximal suspensory as PET is excellent at detecting active proximal suspensory enthesopathy.’   

PET tells us about tissue  activity  - it does not tell us about pain.

Line 246 ‘MRI has the ability to detect early trabecular subchondral bone changes with the detection of fluid signal, but changes in the dense compact bone typically are not recognized until demineralization happen allowing for accumulation of fluid.’

This is misleading – increased signal intensity in STIR images does not necessarily reflect fluid. Demineralisation is not synonymous with fluid accumulation.

Line 321 ‘The availability and cost of the PET radiotracers have for the moment remain

an obstacle to development of equine PET outside of the USA but these limitations will hopefully be overcome with the growth of PET use in human medicine.’

Makes no sense as written – rephrase

All figure legends should make it clear  what the image is e.g., dorsal scintigraphic image; right is on the right. Generally the figure legends are much improved.

There remains inconsistency in the presentation of references - this is sloppy - either abbreviate all journal names with the correct capitalisation, or write them all in full with the correct capitalisation

Comments on the Quality of English Language

minor revisions needed

Author Response

I have carefully re-read the entire manuscript and remain concerned about the scintigraphy section. Any published paper should provide accurate information. It will after all potentially be cited in the future. A review should cite appropriate recent literature, critically discussed in the light of the authors’ experience, even if the authors were invited to in part provide their opinion on the current roles of scintigraphy and PET. There are studies that have critically looked at the usefulness of scintigraphy in various anatomical locations and this paper should include these either to support the authors’ experiences or the authors need to point out in what circumstances their experiences have been different.

The authors point out that in the distal aspect of limbs other imaging techniques have largely superseded scintigraphy, but point out a few instances in which scintigraphy may be valuable – but these are largely not supported by references – and references are available and should be cited. Moreover, the authors should make it clear how scintigraphy can potentially provide useful information in those scenarios.

Likewise in the vertebral column and pelvic regions, and proximal aspects of the limbs there are references demonstrating the clinical usefulness of scintigraphy which have not been included and need to be.

Any discussion of the value of scintigraphy should make it clear that IRU is not synonymous with pain causing lameness or poor performance, and that IRU may persist long-term, so the presence of IRU must be interpreted with care. It may also be influenced by exercise history and work discipline.

I therefore consider that the scintigraphy section needs fairly major revision before this paper could be considered acceptable for publication.

We have provided the requested major revisions, adding many references (total 59 vs 45 previously) and insisting on the limitations of scintigraphy.

Line 50 This would read better as 'In particular, the aim is to illustrate, based on the authors' clinical experience, ....'
Edited

I still feel that it is necessary to provide references to relevant literature to support your clinical observations and to acknowledge where your clinical experience is at variance from the published literature.
We have added further references.

Any published paper is liable to be quoted in the future and there is a danger of perpetuating information which is not based on properly conducted clinical observational studies, but just on remembered observations.
Indeed these are the limitations of “opinion pieces” we have tried to reduce these to the minimum.

Subsection 2 Based on the way this is now titled 'original indications for scintigraphy' you need to clarify in what situations you no longer feel it is relevant - which you do for tarsal scintigraphy & a central tarsal bone fracture but not for fetlock imaging in a racehorse - the reader is left 'dangling' – not knowing what your experience is, nor what the literature tells us.
We have clarified and added references.

Lines 68-69 Reference needed
references added

Line 81 So how are you suggesting that aortoiliacofemoral thrombosis should be diagnosed now?
We added refs about ultrasound

Line 82 'has been attempted'

This is potentially misleading - see Dyson S, Murray R, Schramme M & Branch M. Lameness in 46 horses associated with deep digital flexor tendonitis in the digit: diagnosis confirmed with magnetic resonance imaging. Equine vet. J. 2003, 35: 681 - 690
We edited and cited the provided reference.

Lines 89 & elsewhere ‘pertinent’ is still repeated numerous times
Apologies, we have attempted to rephrase as much as possible

Lines 93-5 More relevant references need to be given because this does not read like personal clinical experience based on facts. If you can provide clinical data that would be useful as well! – supportive of your statements.
We added more references through the text.

Lines 101-3 So what does high specificity but low sensitivity actually imply with respect to usefulness?
We added “suggesting it should not be used as an isolated or indiscriminate tool for assessment of lameness or poor performance”

Line 108 false negative results
Corrected

Line 110 repetition
Corrected

Line 117 ‘but under the right circumstances, acknowledging the risk for false negatives, whole body scintigraphy remains a useful tool

So what are the ‘right circumstances’ ? You cite a 2007 reference - & Archer et al was a review of previous publications , before 2007 - we are now in 2024! – and other publications based on clinical evidence have superseded that! And what about false positive results?
We edited and added references

Line 123 'it appears that a combination of different modalities provides the most comprehensive approach to complex lameness issues ' - but you have not really provided many examples - in the first paragraph you compare scintigraphy with radiography - this is not new!
This was as we were reflecting on original roles of scintigraphy.

Line 135 'Left-right asymmetry in uptake in the lumbar articular facets is an important parameter to consider as a cause or consequence of asymmetrical gait or work.'

Increased radiopharmaceutical uptake does not cause lameness - it may be associated with lameness.
Edited

Please be consistent in nomenclature – the correct term is articular process joints – and the IRU is in the bone – so in the articular processes.
Corrected

Line 138 Imaging of ribs is not challenging – you concurred with this previously yet have not changed the text – this is annoying.
We removed the sentence

Line 142 repetition
Unclear what this refers to, sorry

Lines 146 - 150 but IRU in muscle may be an incidental finding and in some horses with clinically significant myopathy scintigraphy gives false negative results – the information delivered must be accurate.
Edited as suggested

Line 161 'Especially in sport horses a mixture of signal intensities can be seen in degenerative pathologies. ' - what is this supposed to mean?
Removed

Lines 168- 170 ‘Other examples in the distal limb where scintigraphy commonly assists in diagnosis making are the palmar/plantar processes of the distal phalanx, the navicular bone and the sagittal groove of the proximal phalanx.’ How? References needed. And when increased signal intensity in STIR images is the main finding from MRI scintigraphy is usually negative.
We specified this was an observation from one of the authors

Line 176 ‘However regions of increased uptake in sport horses may remain present throughout the career of the horse, given the degenerative nature of many conditions.’ This is an odd statement. IRU associated with an exostosis on McII for example may persist long-term – this is not a degenerative condition. IRU in a tuber ischium may be present long term – this is not a degenerative condition.
We did not imply that all were degenerative conditions.

Line 222 ‘Another important role of PET in the distal tarsal and proximal metatarsal region is to distinguish the origin of pain in the area between distal tarsal joint or proximal suspensory as PET is excellent at detecting active proximal suspensory enthesopathy.’

PET tells us about tissue activity - it does not tell us about pain.
Indeed. We rephrased.

Line 246 ‘MRI has the ability to detect early trabecular subchondral bone changes with the detection of fluid signal, but changes in the dense compact bone typically are not recognized until demineralization happen allowing for accumulation of fluid.’

This is misleading – increased signal intensity in STIR images does not necessarily reflect fluid. Demineralisation is not synonymous with fluid accumulation.
What we are expressing is that demineralization in compact bone is necessary to allow the space for fluid accumulation

Line 321 ‘The availability and cost of the PET radiotracers have for the moment remain

an obstacle to development of equine PET outside of the USA but these limitations will hopefully be overcome with the growth of PET use in human medicine.’

Makes no sense as written – rephrase
We edited

All figure legends should make it clear what the image is e.g., dorsal scintigraphic image; right is on the right. Generally the figure legends are much improved.
The exact location of the image is not necessarily mentioned in the figure title but is available in the legend.

There remains inconsistency in the presentation of references - this is sloppy - either abbreviate all journal names with the correct capitalisation, or write them all in full with the correct capitalization.
All references were made using a reference management software (endnote) after downloading the references from the publisher websites. The capitalization is coming from the journals themselves that at some times were providing the titles in ALL CAPS. We will let the editorial team decide what is the appropriate reference format for this journal.
“Sloppy” is incorrect, unnecessary and disrespectful.